# PRISMA Systematic Literature Review, including with Meta-Analysis vs. Chatbot/GPT (AI) regarding Current Scientific Data on the Main Effects of the Calf Blood Deproteinized Hemoderivative Medicine (Actovegin) in Ischemic Stroke

**DOI:** 10.3390/biomedicines11061623

**Published:** 2023-06-02

**Authors:** Aurelian Anghelescu, Florentina Carmen Firan, Gelu Onose, Constantin Munteanu, Andreea-Iulia Trandafir, Ilinca Ciobanu, Ștefan Gheorghița, Vlad Ciobanu

**Affiliations:** 1Faculty of Midwifery and Nursing, University of Medicine and Pharmacy “Carol Davila”, 020022 Bucharest, Romania; aurelian.anghelescu@umfcd.ro; 2The Neuromuscular Rehabilitation Clinic Division, Teaching Emergency Hospital “Bagdasar-Arseni”, 041915 Bucharest, Romania; andreea-iulia.trandafir@drd.umfcd.ro (A.-I.T.); ciobanuilinca@gmail.com (I.C.); stefangheorghitaa@yahoo.com (Ș.G.); 3The Physical and Rehabilitation Medicine & Balneology Clinic Division—The NeuroRehabilitation Compartment, Teaching Emergency Hospital of the Ilfov County, 22104 Bucharest, Romania; firancarmen@yahoo.com; 4Faculty of Medicine, University of Medicine and Pharmacy “Carol Davila”, 020022 Bucharest, Romania; 5Faculty of Medical Bioengineering, University of Medicine and Pharmacy “Grigore T. Popa” Iași, 700454 Iași, Romania; 6Computer Science Department, Politehnica University of Bucharest, 060042 Bucharest, Romania; vlad.ciobanu@upb.ro

**Keywords:** ischemic stroke, chatbot/GPT (artificial intelligence—AI), deproteinized ultrafiltrate/hemodialysate compound Actovegin, PRISMA-type systematic literature review, meta-analysis

## Abstract

Background: Stroke is a significant public health problem and a leading cause of death and long-term disability worldwide. Several treatments for ischemic stroke have been developed, but these treatments have limited effectiveness. One potential treatment for this condition is Actovegin^®^/AODEJIN, a calf blood deproteinized hemodialysate/ultrafiltrate that has been shown to have pleiotropic/multifactorial and possibly multimodal effects. The actual actions of this medicine are thought to be mediated by its ability to reduce oxidative stress, inflammation, and apoptosis and to enhance neuronal survival and plasticity. Methods: To obtain the most up-to-date information on the effects of Actovegin^®^/AODEJIN in ischemic stroke, we systematically reviewed the literature published in the last two years. This review builds upon our previous systematic literature review published in 2020, which used the Preferred Reporting Items for Systematic Reviews and Meta-Analyses (PRISMA) method to search for and select related articles over almost two decades, between 1 January 2001 and 31 December 2019. Additionally, we compared the results of our PRISMA search (human intelligence-based) with those obtained from an interrogation of a GPT-based chatbot (ChatGPT) in order to ensure comprehensive coverage of potentially relevant studies. Results: Our updated review found limited new evidence on the use of Actovegin^®^/AODEJIN in ischemic stroke, although the number of articles on this subject consistently increased compared to that from our initial systematic literature review. Specifically, we found five articles up to 2020 and eight more until December 2022. While these studies suggest that Actovegin^®^/AODEJIN may have neuroprotective effects in ischemic stroke, further clinical trials are needed to confirm these findings. Consequently, we performed a funnel analysis to evaluate the potential for publication bias. Discussion: Our funnel analysis showed no evidence of publication bias, suggesting that the limited number of studies identified was not due to publication bias but rather due to a lack of research in this area. However, there are limitations when using ChatGPT, particularly in distinguishing between truth and falsehood and determining the appropriateness of interpolation. Nevertheless, AI can provide valuable support in conducting PRISMA-type systematic literature reviews, including meta-analyses. Conclusions: The limited number of studies identified in our review highlights the need for additional research in this area, especially as no available therapeutic agents are capable of curing central nervous system lesions. Any contribution, including that of Actovegin (with consideration of a positive balance between benefits and risks), is worthy of further study and periodic reappraisal. The evolving advancements in AI may play a role in the near future.

## 1. Introduction

Stroke is one of the most critical and life-threatening neurological issues caused by blood circulation disorders in the brain [1]. It presents deep psychosocial and economic challenges [2]. Ischemic stroke, which accounts for approximately 80% of all strokes, is caused by the obstruction of a blood vessel supplying the brain, leading to ischemia and, eventually, cell death. This often triggers a cascade of interconnected events, many of which evolve into vicious cycles resulting in secondary lesions and extended pathological consequences, including long-term impairments. Currently, there are no therapeutic interventions capable of fully restoring central nervous system (CNS) lesions [3]. Therefore, any therapeutic agent, including components of calf blood deproteinized hemodialysate, that has shown promising results or even small functional gains [4] with a positive balance between benefits and risks represents a promising healing approach [2].

The AHA/ASA 2021 Guidelines for the Prevention of Stroke in Patients with Stroke and Transient Ischemic Attack from the American Heart Association/American Stroke Association emphasizes the importance of a comprehensive and individualized approach to stroke prevention. It emphasizes the consideration of each patient’s unique risk profile and clinical characteristics [5]. The management of pediatric stroke poses even greater challenges due to delayed diagnosis, the absence of standardized protocols, and limited clinical trials [6].

Actovegin primarily acts on various aspects of the ischemic stroke lesion constellation, with a focus on tissue oxidation [7], energy metabolism [8], and glucose availability [9]. It elevates these factors, combating ischemic processes and oxidative stress [10]. Additionally, Actovegin decreases inflammation, potentially impacting the neutrophil-to-lymphocyte ratio [11] by modulating the nuclear factor-κB pathway and apoptosis-like processes. It also inhibits caspase-3 activation induced by amyloid β-peptides [12].

In line with the appropriate scientific context, most members of our team conducted and published a systematic review in 2020. The review focused on the pleiotropic effects of Actovegin in relation to the injury pathways of ischemic stroke. Our specific aim was to emphasize Actovegin’s well-known beneficial and therapeutic effects, which either directly or indirectly interfere with the morbidity pathways of ischemic stroke [2]. To achieve this goal, we conducted a comprehensive search of related articles published between 1 January 2001 and 31 December 2019 in reputable international medical databases, including the National Center for Biotechnology Information (NCBI)/PubMed, NCBI/PubMed Central (PMC) (NCBI/PubMed, NCBI/PMC, available online at https://www.ncbi.nlm.nih.gov, accessed on 6 April 2023), Elsevier (available online at https://www.elsevier.co, accessed on 6 April 2023), Physiotherapy Evidence Database (PEDro) (PEDRO Score, available online at https://www.strokengine.ca/glossary/pedro--score, accessed on 6 April 2023), and the Institute for Scientific Information (ISI) Web of Science (Institute for Scientific Information (ISI) Web of Knowledge/Science, available online at https://apps.webofknowledge.com, accessed on 6 April 2023) (via ISI Thomson Reuters). Human intelligence was employed in our search process. The same keywords and search criteria were applied consistently throughout our investigation, including “stroke”, “Actovegin”/”stroke”, “calf blood deproteinized hemodialysate”/“stroke”, “calf blood deproteinized ultrafiltrate”/“stroke”, “Actovegin”, “pleiotropic”/“stroke”, “calf blood deproteinized hemodialysate”, “pleiotropic”/“stroke”, “calf blood deproteinized ultrafiltrate”, and “pleiotropic” [2].

The selection method also involved the Preferred Reporting Items for Systematic Reviews and Meta-Analyses (PRISMA), based on a focused, step-by-step classification according to the primarily used literature identification stages. Within the PRISMA paradigm, we considered only open-access/free full-text articles written in English and indexed in the ISI Thomson Reuters database. There were thirteen reports remaining after ISI checking and duplicates were removed. The final PEDro score for each article considered was calculated via a weighted average formula; lastly, we retained only those articles that obtained a score of at least four, and we found only five articles in the indexed ISI-Reuters database with content that was closest to our query [2].

Previously, we noticed the poverty of bibliographic resources published in the studied period; as we will emphasize further in our present article, the interest in Actovegin’s biological and therapeutic effects seems to have grown significantly in the last two years.

Chatbots are becoming important gateways to digital services and information in domains such as customer service, health, education, and work support. Chatbots are conversational agents that provide access to information. There are several ways chatbots can be involved in scientific research papers [13]. For example, researchers can program chatbots to ask specific questions or follow a particular protocol, which can help ensure data collection consistency [14].

## 2. Methods

Given the previously reached conclusions regarding Actovegin^®^, our goal was to find whether progress was noted regarding Actovegin^®^ use in ischemic stroke in the last three years (2020–2022). Accordingly, we conducted a second systematic literature review regarding the use of Actovegin^®^ in stroke. For this purpose, we searched for and interrogated related articles for the period of 1 January 2020–31 December 2022 in reputable international medical databases: The National Center for Biotechnology Information (NCBI)/PubMed, NCBI/PubMed Central (PMC), Elsevier, Physiotherapy Evidence Database (PEDro), and the Institute for Scientific Information (ISI) Web of Knowledge/Science (via ISI Thomson Reuters index check). In searching, we used specific sets of keywords: “stroke”, “Actovegin”/“stroke”, and “calf blood deproteinized hemodialysate”/“stroke”, “calf blood deproteinized ultrafiltrate”/“stroke”, “Actovegin”, “pleiotropic”/“stroke”, “calf blood deproteinized hemodialysate”, “pleiotropic”/“stroke”, “calf blood deproteinized ultrafiltrate”, and “pleiotropic”. Table 1 shows the numerical results of our search, which were based on a focused, step-by-step classification according to the literature identification and selection stages, and the Preferred Reporting Items for Systematic Reviews and Meta-Analyses (PRISMA) (see Figure 1).

The PRISMA approach was used to identify relevant articles written in English and indexed in the ISI Thomson Reuters database. A custom evaluation algorithm was used to assess each article’s scientific impact and quality based on the year of publication, the total number of citations, and the PEDro score.

Only articles with a score of at least four were included in the analysis. The search identified limited articles related to Actovegin^®^ and its use in stroke treatment, indicating a need for further investigation in this area. Therefore, all of the papers identified by keywords were reviewed, and additional relevant articles were purchased for inclusion in the study.

Notably, we read all of the articles identified by keywords, even if they did not qualify according to our customized PEDro-inspired indirect quality classification (see Table 1). 

Using a mixed-method approach to gathering relevant literature is common in systematic reviews, especially when the topic is relatively narrow and scarce. However, it is essential to note that non-standardized methods may introduce bias in selecting papers and should be used cautiously. To minimize bias, it is recommended to clearly state the inclusion and exclusion criteria and document the search process, including the sources, used search terms, and search dates (see Table 2).

The selected articles were the inputs for a funnel analysis [29]. One could use the “paradigm funnel “as a research tool to produce an enlightened study of complex scientific literature. The literature review is a central building block for academic research. The evidence funnel is a model that emphasizes the responsibility of healthcare professionals in making decisions based on the best available evidence, clinical expertise, and patient values. It reflects the diversity of relevant information sources [30]. The funnel model acknowledges that evidence is gained through different research designs and from various sources, and it is the responsibility of the healthcare professional to make reasonable assessments of the available evidence. Ultimately, the evidence funnel model emphasizes the importance of evidence-based decision-making in healthcare [31] (see Figure 2).

## 3. Chatbot (AI) Interrogation Regarding Actovegin^®^ in Ischemic Stroke

Artificial Intelligence (AI) has been trending excessively in the last year with more and more groundbreaking results. One can play chess with an AI-powered opponent, be guided by one through traffic, or even be driven autonomously through a crowded city. In the medical field, AI tools, such as the Ethereum blockchain, ensure data preservation, decentralization, and immutability [32]. As another example, a preoperative predictive model combining regression and neural network modeling can effectively identify patients at risk of postoperative urinary retention in lumbar surgery [33]. Even though everything looks exciting and sometimes even worrying, we are still far from reaching a strong AI era, a period in which AI is “able to match human intelligence and has the ability to make decisions independently, like humans” [34].

Most of the current successful AI implementations are still trying to break weak AI principles, which are “domain dependent” [35] and mainly able to help in three different methods: scoring, classification, and interpolation or regression. However, one cannot argue with the significant benefits these techniques bring by accelerating and even taking over tasks that humans previously performed, through intensive machine learning algorithms. The latest addition to the world of AI is ChatGPT, a conversational chatbot based on a natural language understanding and processing model developed by OpenAI in 2022 [36]. Technically speaking, ChatGPT is a work of art, being able to do both supervised and unsupervised training; mimic a human chat interaction to a point where it is not clear whether one is talking to a bot or an actual human; and summarize and answer questions on a vast matter of subjects, thus exceeding the normal knowledge levels of a human. 

Our interrogation of ChatGPT is included in Appendix A. The main conclusion was that for general questions (e.g., “*Q1. What is Actovegin?*” or “*Q5. Do you know what PRISMA is?*”), the chatbot can answer with generic but relevant information, showing that it understands its limits (see *Q6: Can you do a systematic review for Actovegin and ischemic stroke?*). However, the main concern comes when asking for more in-depth information where there is no room for guessing or predictions (see “*Q2: What studies are there about Actovegin and ischemic stroke?*”, “*Q3: Can you tell me the authors and titles of the studies conducted on the effect of Actovegin on ischemic stroke?*” *or* “*Q4: Are these the most relevant studies?*”). The disturbing fact comes when checking the articles that ChatGPT provided as “some of the more recent and well-known studies on the topic of Actovegin and ischemic stroke”. Wondering why our PEDro-inspired search mechanism did not find these articles, we tried to find papers to identify why the database searches yielded no results. We unsuccessfully searched for them generically on Google. Next, we identified the contents for each of the given journals except the last one, where the language barrier stopped us from recognizing the online reference (https://www.hindawi.com/journals/srt/contents/year/2017/page/1/; https://www.sciencedirect.com/journal/journal-of-clinical-neuroscience/vol/71/suppl/C
https://www.sciencedirect.com/journal/brain-research-bulletin/vol/125/suppl/C
https://www.strokejournal.org/issue/S1052-3057(17)X0011-6), and we were surprised to see that there were no such articles in the given issues.

## 4. Results Seen as Progress in the Last Three Years Resulting from PRISMA-Type Systematic Review

Actovegin is a deproteinized hemoderivative obtained from calf blood that is processed through two stages of ultrafiltration; this results in the removal of all proteins, antigens, and other high-molecular-weight components [27]. The final product is a complex of more than 200 low-molecular-weight bioactive substances with a molecular weight of less than 5000 Da [2,36,37]. These substances include amino acids, oligopeptides, nucleosides, organic acids, and electrolytes. Actovegin^®^ is manufactured by Nycomed Austria GmbH, and it has been approved for use in several countries, including Austria, Germany, Switzerland, Russia, and some others [38], whereas AODEJIN is produced by Avanc Pharmaceutical Co., Ltd., Jinzhou, China.

Actovegin^®^ has been used with various medical indications for over 60 years, including stroke or cartilage degeneration [28]. Our previous related article detailed its pleiotropic, multifactorial, and possibly multimodal actions [2].

Recently, the effect of Actovegin^®^ on the release of inflammatory cytokines IL-1β, IL-6, IL-10, and TNF-α was investigated, revealing its potential to support antioxidative systems in cells. Furthermore, Actovegin^®^ demonstrated a dose-dependent reduction in the release of pro-inflammatory interleukin Il-1 β in human peripheral blood mononuclear cells, potentially due to a specific effect on B cells [9].

Furthermore, Actovegin^®^ showed anti-inflammatory properties by reducing lipopolysaccharides and Phorbol 12-myristate 13-acetate (PMC)-induced reactive oxygen species (ROS), which may contribute to the formation and function of an efficient antioxidative and/or anti-inflammatory system in cells, thus reducing the consequences of inflammation. These findings may help in understanding the positive effects of Actovegin on inflammatory lesions [9].

Actovegin (AODEJIN) has been reported to have metabolic effects that could potentially enhance the energy provided to the brain at the cellular level after ischemia or trauma. In addition, Actovegin has been shown in a preclinical study to inhibit apoptosis, thus enhancing cell survival, improving spatial learning and memory, and reducing oxidative stress [10].

The safety of Actovegin has not been extensively evaluated in studies with a control group and long-term follow-up, despite its widespread usage in many countries over several decades. Of the five studies included in the systematic review, one study was designed to collect prospective data on adverse events with Actovegin compared to a placebo; it reported a numerically higher discontinuation rate due to adverse events. This study also reported a higher incidence of recurrent ischemic stroke, transient ischemic attack, or intracerebral hemorrhage in patients taking Actovegin compared to a placebo [27]. However, other studies that evaluated the role of Actovegin in the care of adult patients after an ischemic stroke did not identify significant improvement in mortality or morbidity with the use of Actovegin. Therefore, the currently available data that consider the benefits of Actovegin for patients with ischemic stroke are uncertain, with a potential risk of harm [26,28]. More evidence is needed from rigorously designed clinical trials to justify the role of Actovegin, including the balance between benefits and risks in patients with ischemic stroke.

## 5. Meta-Analysis

The PICOT methodology was applied to select data about using Actovegin in all available completed clinical trials. This methodology included the following criteria: Population: individuals with ischemic stroke; Intervention: Actovegin treatment; Comparison: placebo or standard treatment; Outcome: improvement in neurological function or other measures of stroke recovery; Time frame: studies published up until the present day. Using these PICOT criteria, we searched for all clinical trials that were relevant to our meta-analysis (see Table 3).

The meta-analysis included only two studies, ARTEMIDA and APOLLO (see Figure 3). The total number of included patients with stroke was 869. A forest plot indicated small advantages for Actovegin compared to a placebo. Distinct from the above-mentioned study [9], which could not be due to a related meta-analysis because of the considerable heterogeneity of the analyzed clinical trials, we proceeded with such an endeavor. However, based on only two eligible studies, we previously carried out a funnel plot type of evaluation with the purpose of determining possible publication bias. Funnel plots are commonly used in meta-analyses to investigate the presence of publication bias; a deflection/tilt occurs when the published literature is not representative of all the studies conducted on a particular topic. A funnel plot typically indicates the estimated effect from individual studies against their standard errors or some measure of precision, such as the sample size or inverse of the standard error (see Figure 4). In a meta-analysis, a symmetric funnel plot suggests no publication bias, while an asymmetric funnel plot may indicate publication bias or other sources of bias that affect small studies more than large ones. Based on the information obtained, the funnel plot in our meta-analysis may have demonstrated, through symmetry, that there were no publication biases.

## 6. Discussion

Actovegin has been approved for use in many countries, including European countries such as Germany, Austria, and Switzerland (available online at https://www.ema.europa.eu/en/human-regulatory/post-authorisation/data-medicines-iso-idmp-standards/public-data-article-57-database, accessed on 6 April 2023), as well as in Asia and Latin America. In the United States, Actovegin has not been approved by the FDA for any medical use (available online at https://www.fda.gov/search?s=Actovegin, accessed on 6 April 2023). However, it can be imported for personal use under certain conditions, such as treating a rare disease.

The use of multimodal medications [40] for treating ischemic stroke has to be discussed. For such a complex pathology as ischemic stroke, using medicine with only one mechanism of action is neither ideal nor effective, and data suggest that medications with pleiotropic or multimodal actions [12] may produce better outcomes. There are subtle and complex interferences and overlaps between the various endogenous neurobiological processes involved in the treatment of ischemic stroke, including neuroprotection, neurotrophicity, neuroplasticity, and neurogenesis. Note that these processes are intermingled with the effects of deproteinized calf blood medicine (Actovegin^®^), and evolve over weeks, months, and potentially years, with rehabilitation and neural repair as critical factors in the subacute and chronic phases of treatment. Our review emphasizes the need to better understand the complex interactions between these processes in order to optimize treatment outcomes.

Chatbots are artificial intelligence (AI) programs that use natural language processing (NLP) to simulate human conversations. Chatbots can answer questions, provide information, and perform tasks, including those that involve scientific knowledge. However, chatbots are limited by their programming, and may not be able to offer the same level of detail or accuracy as a systematic review methodology such as PRISMA. In the context of comparing PRISMA and chatbots’ capabilities for reviewing scientific knowledge related to Actovegin in ischemic stroke, it is essential to note that PRISMA is a rigorous methodology for conducting systematic reviews of scientific literature. In contrast, chatbots may be more suited to providing quick and simple answers to specific questions. While chatbots can provide a broad overview of the current scientific knowledge of Actovegin in ischemic stroke, a systematic review using PRISMA would provide a more thorough and detailed analysis of the available evidence.

The GPT-based chatbot is an AI tool that produces text resembling human writing; it allows users to interact with AI almost as if they are communicating with another person. For instance, if asked to designate specifically related studies, ”the software will provide one, even if nobody has written one before” [41].

Recent studies suggest that while chatbots may serve as a low-threshold interface to information, services, and societal participation, they may also face challenges regarding bias and inclusion. Moreover, there has been a lack of more systematic or structured investigations regarding chatbots’ universal and inclusive design. Chatbots’ inclusive and responsible design requires an understanding of various linguistic elements of conversation and contextual factors [41].

However, there are some limitations to using chatbots in scientific research papers. One major limitation is the potential for bias in chatbot interactions. Chatbots rely on natural language processing algorithms to understand and respond to user input. These algorithms can be influenced by factors such as how questions are phrased or the tone of voice used by the chatbot. Additionally, chatbots may be unable to pick up on nonverbal cues or emotions, which could impact the accuracy of the data collected. To address these challenges, there is a need for more systematic investigations of the universal and inclusive design of chatbots. The inclusive and responsible design of chatbots requires an understanding of various linguistic elements of conversation and contextual factors [42]. 

One possible way that chatbots can assist in this process is by automating the screening of articles for inclusion/exclusion criteria. For example, in a study by Okonkwo et al. (2021), a chatbot was developed to screen articles for inclusion in a systematic review using natural language processing techniques [43]. Regarding a parallel between PRISMA systematic reviews and chatbot output, both involve systematically screening and extracting information from articles. However, while PRISMA involves a manual process of article screening and data extraction, chatbots can automate these processes using natural language processing techniques. This can potentially reduce the time and effort required for conducting systematic reviews. In conclusion, chatbots have the potential to assist researchers in conducting systematic reviews and writing scientific articles. However, further research is needed to develop and refine chatbot technologies for these purposes and to ensure their accuracy and reliability.

## 7. Conclusions

In this article, we presented a comparison between a systematic literature review using the PRISMA method—performed by human intelligence—and an AI-based chatbot (ChatGPT) in order to gather current information on the use of Actovegin in ischemic stroke. The systematic literature review found limited and conflicting evidence on the efficacy and safety of Actovegin in ischemic stroke, with some studies showing no significant benefit over a placebo while others reported a higher rate of good functional outcomes and lower incidences of recurrent stroke and mortality; this was slightly shown overall in our meta-analysis. On the other hand, the AI-based chatbot provided general information about Actovegin and its use in ischemic stroke. Still, it could not critically evaluate the quality of evidence, provide a comprehensive analysis of the literature, or provide actual and beyond-question data. On the one hand, our ChatGPT interrogation generally received fluent and coherent answers, including fair ones regarding its current lack of capabilities to achieve systematic literature reviews and meta-analyses of the PRISMA kind; on the other hand, it provided some bibliographic resources we could not find either within our standardized literature search or in open sources.

Since AI can produce authoritative-sounding output that may be inaccurate, incomplete, or biased, the technology should be applied under human supervision and control, and authors should carefully review and edit the results. 

Overall, our review highlights the importance of using rigorous systematic review methods for evaluating medical treatments, and suggests that AI-based tools may have limitations in providing a complete and accurate picture of the available evidence, to say nothing about the significant discussions regarding the risks of a “Sorcerer’s Apprentice” kind of behavior in the advancement of AI technology.

## Figures and Tables

**Figure 1 biomedicines-11-01623-f001:**
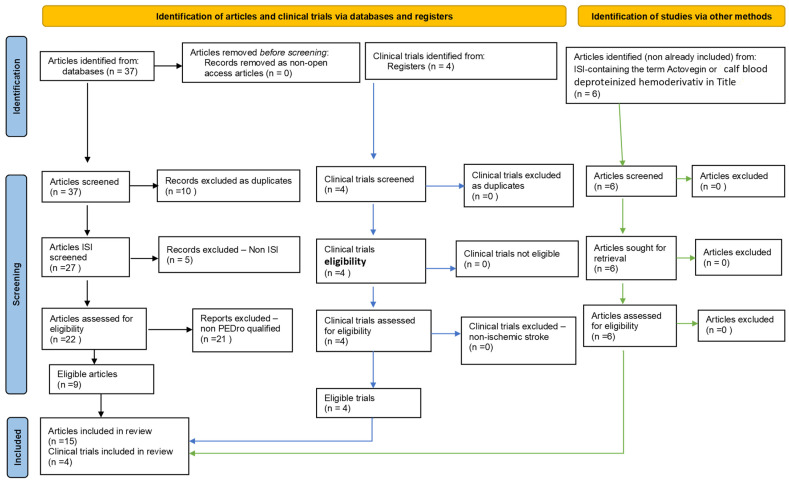
Adapted from a Preferred Reporting Items for Systematic Reviews and Meta-Analyses (PRISMA) flow diagram, customized for our study [15].

**Figure 2 biomedicines-11-01623-f002:**
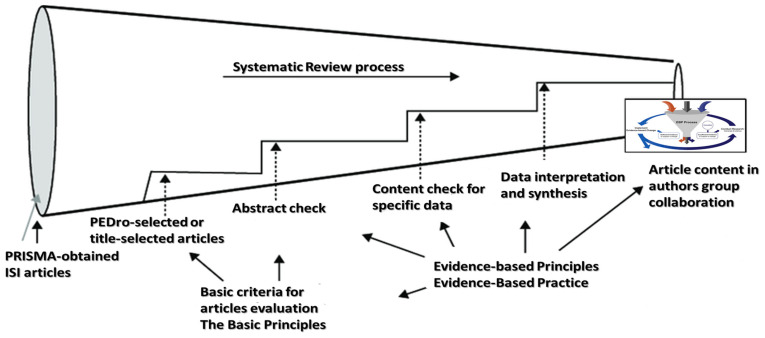
An adapted form of the funnel analysis flow diagram, customized for our study [15].

**Figure 3 biomedicines-11-01623-f003:**
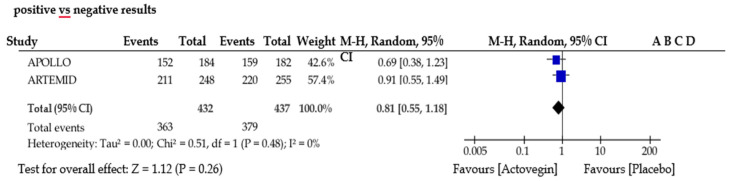
Actovegin for Ischemic Stroke Meta-Analisys—positive vs. negative results.

**Figure 4 biomedicines-11-01623-f004:**
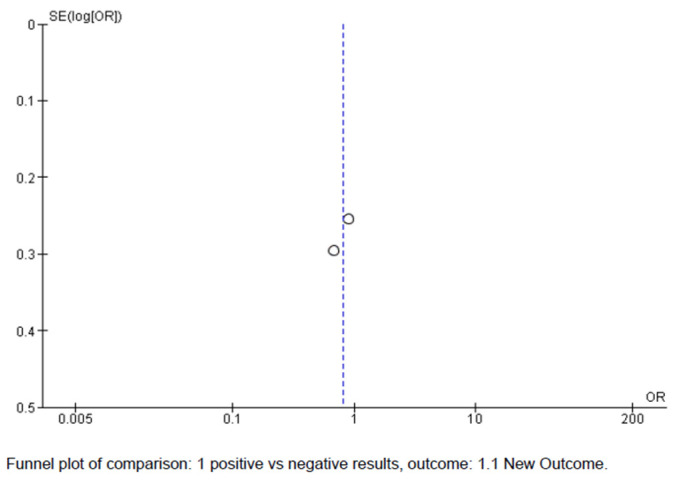
Funnel plot—risk of bias.

**Table 1 biomedicines-11-01623-t001:** Keyword set-based search for related articles—numerical results. PMC: PubMed Central and PEDro: Physiotherapy Evidence Database.

Keywords	Elsevier	PubMed	PMC	PEDro	Total
“stroke” + “Actovegin”	0	3	27	0	30
“stroke” + “calf blood deproteinized hemoderivative”	0	0	0	0	0
“stroke” + “calf blood deproteinized hemodialysate”	0	0	0	0	0
“Actovegin” + “pleiotropic”	0	0	7	0	7
“calf blood deproteinized hemoderivative” + “pleiotropic”	0	0	0	0	0
“calf blood deproteinized hemodialysate” + “pleiotropic”	0	0	0	0	0
Total	0	3	34	0	37

**Table 2 biomedicines-11-01623-t002:** ISI–Reuters database-indexed articles obtained and included in the systematic review.

Reference	Keywords	Publication_Year	Isi_Citation_Count	References_Count	PEDRO_Score
[16]	“stroke” + “Actovegin”	2020	66	413	10
[17]	“stroke” + “Actovegin”	2021	30	165	10
[18]	“stroke” + “Actovegin”	2021	30	46	10
[19]	“stroke” + “Actovegin”	2020	17	69	9
[20]	“stroke” + “Actovegin”	2021	12	22	6
[21]	“stroke” + “Actovegin”	2021	10	34	5
[22]	“stroke” + “Actovegin”	2020	10	42	5
[2]	“stroke” + “Actovegin”	2020	8	75	4
[23]	“stroke” + “Actovegin”	2020	7	48	4
[24]	“stroke” + “Actovegin”	2020	3	36	2
[25]	“stroke” + “Actovegin”	2022	3	146	2
[10]	“stroke” + “Actovegin”	2021	2	38	1
[9]	“stroke” + “Actovegin”	2020	2	75	1
[26]	“stroke” + “Actovegin”	2021	2	37	1
[27]	“stroke” + “Actovegin”	2022	0	31	0
[28]	“stroke” + “Actovegin”	2022	0	26	0

**Table 3 biomedicines-11-01623-t003:** Clinical Studies identified on ClinicalTrials.gov.

Ref.	Study Name/ID	Publication Year	Population	Actovegin Treatment	Placebo	Outcome
[8]	ARTEMIDA Unique identifier: NCT01582854.	2017	503	248	255	Actovegin had a beneficial effect on cognitive outcomes in patients with post-stroke cognitive impairment. The safety experience was consistent with the known safety and tolerability profile of the drug. These results warrant confirmation in additional robustly designed studies.
[39]	(APOLLO) NCT03469349	2020	366	184	182	The results of this 12-week course of Actovegin demonstrated its superiority over a placebo in the increase in ICD and ACD at weeks 2, 12, and 24 from the start of treatment. Actovegin has an acceptable safety and tolerability profile.
[20]	European Stroke Organisation and European Academy of Neurology joint guidelines on post-stroke cognitive impairment	2021	Includes data only from ARTEMIDA			A beneficial effect of Actovegin compared to a placebo was reported, but the effect size described may be less than the minimal clinically significant difference.
[27]	Actovegin in the management of patients after ischemic stroke: A systematic review	2022	Includes data only from ARTEMIDA and other heterogeneous data			The benefits of Actovegin are uncertain, and there is a potential risk of harm in patients with stroke.

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
