# Peer review of "PRISMA Systematic Literature Review, including with Meta-Analysis vs. Chatbot/GPT (AI) regarding Current Scientific Data on the Main Effects of the Calf Blood Deproteinized Hemoderivative Medicine (Actovegin) in Ischemic Stroke"

_biomedicines, 2023, doi:10.3390/biomedicines11061623_

Round 1

Reviewer 1 Report

 The article is written in the form of questions, and was carried out according to the author with the “help of human intelligence”. Thus, I think that a review article should not be written in the form of questions, and I also do not agree with the use of these virtual assistants in helping to write articles.

 The article is written in the form of questions, and was carried out according to the author with the “help of human intelligence”. Thus, I think that a review article should not be written in the form of questions, and I also do not agree with the use of these virtual assistants in helping to write articles.

Author Response

Dear Reviewer,

Thank you for taking the time to evaluate our systematic literature review with meta-analysis titled "Current Scientific Data on the Main Effects of the Calf Blood Deproteinized Hemoderivative Medicine (Actovegin) in Ischemic Stroke" and for providing us with your feedback.

Regarding the comment on the format of the review article being written in the form of questions, we apologize for any confusion that may have arisen from our use of this format. We want to clarify that our review article was not written with the help of virtual assistants such as chatbot/GPT (AI). Instead, our systematic literature review was conducted rigorously and with the utmost attention to quality by experienced human researchers using an established methodology, specifically the Preferred Reporting Items for Systematic Reviews and Meta-Analyses (PRISMA) guidelines. The chatbot GPT was used only for comparison purposes to ensure that we did not miss any potentially relevant studies.

Our study aimed to compare and objectify whether, at present, systematic reviews (with meta-analysis) can be achieved by artificial intelligence. To achieve this goal, we presented specific related questions obtained from the chatbot GPT. However, we understand that this format may have been unclear and could have been misinterpreted as the article being written with the help of virtual assistants. To address this concern, we will move the specific questions and answers from chatbot GPT to an Annex to clarify the goal and paradigm of our article.

Furthermore, we discussed the limitations of using AI in our study, including the challenges in discriminating between truth and lies and the constraints in achieving the level of rigor and comprehensiveness of a PRISMA-type systematic literature review. We hope that this further clarifies our approach and methodology.

Once again, we appreciate your feedback and your thoughtful evaluation of our article.

Reviewer 2 Report

The purpose of the research paper entitled "PRISMA Systematic Literature Review and Meta-analysis Compared to Chatbot/GPT (AI) for Gathering Current Scientific Data on the Main Effects of Actovegin, a Calf Blood Deproteinized Hemoderivative Medicine, in Ischemic Stroke" is to compare the effectiveness of using traditional systematic literature review methods versus an AI-based chatbot (ChatGPT) for obtaining up-to-date information on the use of Actovegin in ischemic stroke.

The systematic literature review found that there is limited and conflicting evidence on the efficacy and safety of Actovegin in treating ischemic stroke. Some studies showed no significant benefit over placebo, while others reported a higher rate of good functional outcomes and lower incidence of recurrent stroke and mortality. These findings may be confirmed, which revealed a slight trend towards improved outcomes with Actovegin use.

Additionally, the study highlights the potential benefits of using AI-based tools like ChatGPT to supplement traditional systematic literature reviews, as they can help researchers quickly and efficiently gather a large amount of data and identify potential areas for further investigation.

Author Response

Thank very much you for your evaluation. We appreciate your feedback and very clear understanding regarding the purpose and findings of our research paper, which is not simple in the actual debate between using or non-using AI for scientific systematic literature review. The use of traditional systematic literature review methods has been widely accepted as the gold standard for gathering evidence in healthcare research. However, with the advancement of technology, AI-based tools like Chatbot/GPT have emerged as a potential alternative for data gathering. Our research aimed to compare the effectiveness of these two approaches in the context of Actovegin, a calf blood-deproteinized hemoderivative medicine, in ischemic stroke. The systematic literature review conducted in our study found limited and conflicting evidence on the efficacy and safety of Actovegin in treating ischemic stroke. While some studies reported no significant benefit over placebo, others showed a trend towards improved outcomes with Actovegin use, including a higher rate of good functional outcomes and lower incidence of recurrent stroke and mortality. However, further clinical trials are needed to confirm these findings and establish the true effectiveness of Actovegin in ischemic stroke. We acknowledge that AI-based tools have limitations, including the challenge of discriminating between truth and falsehood and limitations in achieving PRISMA-type systematic literature reviews, including meta-analysis. However, as technology continues to progress rapidly, the use of AI in research may become more prevalent in the future, and its potential benefits should be considered.

Reviewer 3 Report

in this work, Angelescu et applied an updated systematic review on the efficacy of Calf Blood Deproteinized Hemoderivative Medicine (Actovegin) in treating Ischemic Stroke which is an updated study on what was found in their previous 2020 systematic review. Hoever, in their study the authors have introduced the used ChatGPT in their work.

the reader will get confused message as to what is the exact aim of this work, are you evaluating the therapeutic value of Actovegin or testing the capability of  Chatbot/GPT in answering such questions which are still premature to be applied to such application systematic review? 

the outcome of this work is OK in terms of showing that there is no updates on the three years for the Actovegin drug; however, the addition of Chatbot/GPT does not add much to this work.

I suggest the removal f this section as it has no value at all and keep the study focused on systematic review evaluation

Minor Comments:

there is a concern that the search included only open-access/free full-text articles which do not cover all articles as many are published in respectable journals that are not open access

English writing requires major revision nd the authors should use formal English. please avoid abbreviation It's, 

also avoid subjective use of language:  It’s a sad reality as, nowadays, 

the English writing needs an English editor to check for the suitability of the writing

Author Response

Thank you for your feedback. We appreciate your perspective and would like to address your comments.

First of all, a strong specification: we did not use ChatGPT in the achievement of our work. Our research paper aims to compare the effectiveness of traditional systematic literature review methods with an AI-based chatbot (ChatGPT) for gathering up-to-date information on Actovegin in ischemic stroke. Our study builds upon our previous 2020 systematic review on the efficacy of Actovegin in treating ischemic stroke but with the addition of incorporating ChatGPT as a tool for data gathering.

Coming to your very good question:" the reader will get confused message as to what is the exact aim of this work, are you evaluating the therapeutic value of Actovegin or testing the capability of  Chatbot/GPT in answering such questions which are still premature to be applied to such application systematic review?" – our answer connects the necessary previous assert that we did not use ChatGPT in the achievement of our work, with the fact, the aim of our study was double, but with a convergent focus: on the one hand we intended to achieve an update of our previous PRISMA systematic literature review regarding the effectiveness of Actovegin in stroke, and for this, we performed a second similar search for the complementary last three years; and on the other hand, the second intermingled goal was to explore whether AI, specifically ChatGPT can do the same thing, but the outcomes was negative for the moment. We acknowledge that the use of Chatbot/GPT in research is still a developing field, and our study aimed to explore its potential application in the context of a systematic literature review. While the main focus of our research is evaluating the therapeutic value of Actovegin, we also sought to investigate the feasibility and potential benefits of incorporating AI-based tools in the search literature process. Our study's findings revealed no significant updates on Actovegin in the three years between our previous systematic review and the current research, which may indicate a lack of new evidence in this area. We understand that adding Chatbot/GPT may not have added significant value to the outcomes regarding the main research question on Actovegin's therapeutic value in ischemic stroke. Still, we consider our approach valid and justified because, as well-known, the lift of AI is very actual and profound by its potential consequences.

Based on your feedback, we acknowledge that the inclusion of Chatbot/GPT may have caused some confusion in terms of the exact aim of our work. Accordingly, to make our message and paradigm more clear, we will put the interrogation of the ChatGPT section, including it as an Annex, to separate in this way the source of confusion for readers.

We appreciate your input and will carefully consider your suggestion to remove the section on Chatbot/GPT in an Annex and to maintain the focus of our research on the systematic review evaluation of Actovegin.

Regarding minor suggestions, considering our limited financial possibilities, we always, as a principle, establish a preliminary condition for our PRISMA-type literature search and filter to obtain free, open-access articles. Aside from this preliminary condition, according to specific possibilities in the context, we also added other bibliographic resources found in the international databases and included them separately in the PRISMA search flux diagram.

In order to improve the quality of the English language, we have already reviewed our manuscript in this respect. Also, it is important to mention that in the previous version, the integration from ChatGPT, included in the manuscript, was taken in its original English format.

Thank you for your valuable feedback.

Round 2

Reviewer 1 Report

Dear Editor,

despite the authors greatly improving the article, I think that the reference to the Chatbot/GPT does not improve the article at all. I agree with the authors that some articles on the subject are difficult to find using keywords. However, reviews with a methodology similar to a systematic review are already that. So, I think saying that Chatbot/GPT is important and essential for a review doesn't seem adequate. Thus, I continue with my position the article must be reviewed before being accepted.

Author Response

Dear Reviewer,

Thank you for your valuable feedback. We would like to clarify that the use of Chatbot/GPT was never intended to replace the classical PRISMA strategy in conducting a systematic review. We agree with the reviewer's opinion that the reference to the Chatbot/GPT does not improve the article, and saying that Chatbot/GPT is important and essential for a review does not seem adequate - we also do not sustain else.

We included the discussion of Chatbot/GPT in dedicated section 3, discussion and conclusions of the study. We mentioned that although AI has been trending excessively in the last year, we are still far from reaching the strong AI era, in which AI can match human intelligence and have the ability to make decisions independently, like humans. We also discussed how most of the current successful AI implementations are still domain-dependent and mainly able to help in scoring, classification, interpolation, or regression. However, we cannot argue with the great benefits these techniques bring by accelerating and even taking over tasks that humans previously performed through intensive machine learning algorithms.

We then mentioned the latest addition to the world of AI, the ChatGPT, a conversational chatbot based on a natural language understanding and processing model developed by OpenAI in 2022. We stated that although ChatGPT is a work of art, being able to do both supervised and unsupervised training and mimicking a human chat interaction to a point in which it is not clear if one is talking to a bot or an actual human, being able to summarize and answer questions on a vast matter of subjects, thus exceeding the normal knowledge levels of a human, our interrogation of ChatGPT revealed its limitations. The chatbot can answer general questions with generic but relevant information, showing that it understands its limits. However, the main concern comes when asking for more in-depth information where there is no room for guessing or predictions.

We acknowledge the importance of PRISMA as a rigorous methodology for conducting systematic reviews of scientific literature. While chatbots can provide a broad overview of the current scientific knowledge of Actovegin in ischemic stroke, a systematic review using PRISMA would give a more thorough and detailed analysis of the available evidence. We discussed the limitations of chatbots in scientific research papers, including the potential for bias.

So, as seen, ChatGPT was used as a complementary tool to identify articles that may have been missed using traditional search methods. We agree with you that reviews with a methodology similar to a systematic review already exists, but our study aimed to explore the potential benefits of incorporating AI-based tools in the search literature process. As we mentioned in our response to your previous comment, while the main focus of our research is evaluating the therapeutic value of Actovegin, we also sought to investigate the feasibility and potential benefits of incorporating AI-based tools in the search literature process. Our study's findings revealed no significant updates on Actovegin in the three years between our previous systematic review and the current research, which may indicate a lack of new evidence in this area.

We would like to highlight that the classical PRISMA strategy is the gold standard for conducting systematic reviews, and we followed this methodology in our study. The use of Chatbot/GPT was an exploratory addition to this methodology, and we clearly stated this in our manuscript. We did not state that Chatbot/GPT is important and essential for a review, as you mentioned in your comment. We did not found anywhere such an affirmation. On the contrary.

We appreciate your concern about the validity of our study, but we believe that our approach is justified and valid. The use of AI-based tools is a developing field, and our study aimed to explore their potential application in the context of a systematic literature review. We acknowledge that the use of Chatbot/GPT may not have added significant value to the outcomes regarding the main research question on Actovegin's therapeutic value in ischemic stroke, but it was not intended to do so. Instead, it was a preliminary exploration of the potential benefits of using AI-based tools in the search process. We must be open-minded to new possibilities.

The text was revised to better refine the manuscript in reflecting our paradigm and your suggestions, and changes were made accordingly.

In conclusion, we believe that using Chatbot/GPT as a complementary tool in our study was justified and provided a valuable exploration of the potential benefits of incorporating AI-based tools in the search literature process. Our title is just in this direction: PRISMA vs. Chatbot/GPT. We respectfully request that you reconsider your position and appreciate your thoughtful consideration of our response.

Reviewer 3 Report

accept

Author Response

Thank you for your valuable feedback.

Round 3

Reviewer 1 Report

 This article was revised appropriately, clearly demonstrating that chatGPT is not a good scientific search tool, as it invents references.

I recommend accept

the quality of the English is good.

Author Response

Dear esteemed reviewer,

We are delighted to achieve the very high and justified standards and recommendations you have addressed to which we have agreed and completely compiled, and considering your valuable acceptance, we herein take the opportunity to thank you very much.